# The Competitive Endogenous RNA (ceRNA) Regulation in Porcine Alveolar Macrophages (3D4/21) Infected by Swine Influenza Virus (H1N1 and H3N2)

**DOI:** 10.3390/ijms23031875

**Published:** 2022-02-07

**Authors:** Chao-Hui Dai, Zhong-Cheng Gao, Jin-Hua Cheng, Li Yang, Zheng-Chang Wu, Sheng-Long Wu, Wen-Bin Bao

**Affiliations:** 1College of Animal Science and Technology, Yangzhou University, Yangzhou 225000, China; chdai1993@163.com (C.-H.D.); m15951788188_1@163.com (Z.-C.G.); yangli19970316@163.com (L.Y.); zcwu@yzu.edu.cn (Z.-C.W.); slwu@yzu.edu.cn (S.-L.W.); 2Institute of Animal Science, Jiangsu Academy of Agricultural Sciences, Nanjing 210014, China; jhcheng@jaas.ac.cn; 3Joint International Research Laboratory of Agriculture and Agri-Product Safety, Yangzhou University, The Ministry of Education of China, Yangzhou 225000, China

**Keywords:** pig, influenza virus, porcine alveolar macrophage cell, whole transcriptome sequencing, ceRNA regulation

## Abstract

H1N1 and H3N2 are the two most common subtypes of swine influenza virus (SIV). They not only endanger the pig industry, but are also a huge risk of zoonotic diseases. However, the molecular mechanism and regulatory network of pigs (hosts) against influenza virus infection are still unclear. In this study, porcine alveolar macrophage cell (3D4/21) models infected by swine influenza virus (H1N1 and H3N2) were constructed. The expression profiles of miRNAs, mRNAs, lncRNAs and circRNAs after H1N1 and H3N2 infected 3D4/21 cells were revealed in this study. Then, two ceRNAs (TCONS_00166432-miR10391-MAN2A1 and novel_circ_0004733-miR10391-MAN2A1) that regulated H1N1 and H3N2 infection in 3D4/21 cells were verified by the methods of bioinformatics analysis, gene overexpression, gene interference, real-time quantitative PCR (qPCR), dual luciferase activity assay and RNA immunoprecipitation (RIP). In addition, the important candidate molecules (miR-10391, TCONS_00166432, and novel_circ_0004733) were identified by qPCR and enzyme linked immunosorbent assay (ELISA). Finally, the regulatory effect and possible molecular mechanism of the target gene *MAN2A1* were identified by the methods of gene interference, qPCR, Western blot and ELISA. The results of this study suggested that TCONS_00166432 and novel_circ_0004733 could competitively bind miR-10391 to target the *MAN2A1* gene to regulate swine influenza virus infecting 3D4/21 cells. This study reported for the first time the ceRNA networks involved in the regulation of the swine influenza virus infecting 3D4/21 cells, which provided a new insight into the molecular mechanism of 3D4/21 cells against swine influenza virus infection.

## 1. Introduction

Swine influenza virus (SIV) is an orthomyxovirus with a short incubation period and will spread to the whole herd soon. The infection rate of SIV can be as high as 100% and it can cause concurrent or secondary infections with porcine pleuropneumonia, swine streptococcus disease and porcine reproductive and respiratory syndrome, which makes the condition more complicated and worse and ultimately leads to a sharp increase in mortality in the pig herd [1,2]. Pigs are often regarded as “mixing vessels” of influenza A, which can help to change and develop disease strains and then spread them to other mammals, such as humans [3]. A recent report pointed out that a new SIV strain (G4), similar to the 2009 pandemic virus, could bind to human receptors and produce higher titers of progeny viruses [4]. Therefore, the SIV not only endangers the pig industry, but also is a huge risk of zoonotic diseases. However, the molecular mechanism and regulatory network of SIV infection in host are still unclear. In addition, although the current research reports on swine influenza vaccines, the effectiveness and cross-protection effect cannot be guaranteed [5]. Therefore, revealing the molecular mechanism of pigs against influenza virus infection from the genetic nature can provide important guidance and basis for improving the disease resistance of pigs through molecular breeding in the future.

The whole transcriptome sequencing is an extended RNA-seq technology with constructing two libraries—a small RNA library and a ribosomal chain-specific library, from which information on four types of RNA-microRNA (miRNA), long non-coding RNA (lncRNA), messenger RNA (mRNA) and circular RNA (circRNA) can be analyzed simultaneously. Competing endogenous RNA (ceRNA) reveals a new mechanism for RNA interaction. It is known that microRNA can cause gene silencing by binding to mRNA, and ceRNA can regulate gene expression by competitively binding to microRNA. Studies revealed that lncRNAs and circRNAs could regulate the expression of mRNA (with the same miRNA binding sites) by functioning as ceRNAs (miRNA sponge), namely lncRNA-miRNA-mRNA and circRNA-miRNA-mRNA regulation network [6,7,8]. The whole transcriptome sequencing technology has been widely studied and applied in pigs. Wang et al. [9] constructed the ncRNA-miRNA-mRNA network in Huainan pig muscle and concluded that circRNA might promote fat deposition by adsorbing miR-874 to release the inhibitory effect of miR-874 on *PPARD* gene. Zeng et al. [10] studied lncRNAs and circRNAs in sow milk exosomes and found that some lncRNAs interacted with proliferation-related miRNAs, and some circRNAs might target many miRNAs related to the intestinal barrier. Brogaard et al. [11] analyzed the miRNA expression profile in blood samples before and after influenza A virus (IAV) infection (1, 3, and 14 days) and found that the target genes of regulated miRNAs were involved in apoptosis and cell cycle regulation, which may affect the host response to secondary infection. However, there are few reports about the miRNAs, lncRNAs and circRNAs research in SIV infection. Therefore, the whole transcriptome sequencing was conducted in this study to identify the differentially expressed miRNAs, mRNAs, lncRNAs and circRNAs after SIV (H1N1 and H3N2) infected 3D4/21 cells. This study aims to reveal the expression profiles of four RNAs in the process of 3D4/21 cells against SIV infection for the first time, which could provide a basis for studying transcriptional regulation of H1N1 and H3N2 infecting 3D4/21 cells and provide a theoretical basis for further study of its possible molecular mechanism.

## 2. Results

### 2.1. Establishment of Alveolar Macrophage Cell (3D4/21) Model Infected by H1N1 and H3N2 

After MDCK cells were infected by H1N1 and H3N2, the cytopathic effect (CPE) was calculated separately and the corresponding 50% tissue culture infective dose (TCID_50_) was calculated according to the Reed-Muench method [12]. In this study, the TCID_50_ of H1N1 was 10^−4.7^/100 μL, and the TCID_50_ of H3N2 was 10^−4.7^/100 μL. Viruses (H1N1 and H3N2) of different doses (1-fold TCID_50_, 10-fold TCID_50_ and 100-fold TCID_50_) were used to infect 3D4/21 cells for different times (24 h, 48 h and 72 h). Then the relative expression levels of viral genes (*M* and *NP*) and host cell genes (*RIG-I*, *TLR7* and *NLRP3*) and the secretion levels of cytokines in the cell supernatant were detected. The qPCR results showed that (Figure 1A–D) at the same time point, the expression levels of all genes in 3D4/21 cells infected by H1N1 and H3N2 with 100-fold TCID_50_ were higher than that with 10-fold and 1-fold TCID_50_. For the same virus dose, the expression levels of all genes increased with time increasing, and reached significant levels at 48 h (*p <* 0.05 or *p <* 0.01). There were also significant changes at 72 h, but the difference was not significant compared with 48 h (*p* > 0.05).

The results of ELISA showed that (Figure 1E,F) compared with the cells in the virus-free group (0-fold TCID_50_), the secretion levels of all three cytokines increased significantly after virus infection (*p <* 0.05 or *p <* 0.01). All three cytokines increased at 48 h and decreased at 72 h after cells were infected by virus with 100-fold TCID_50_. In addition, we found that (Figure 1G) the cells showed obvious lesions after infected by H1N1 and H3N2 with 100-fold TCID_50_ for 48 h, and cells began to die and fall off after 72 h. Finally, we found that (Figure 1H) the expression of *NP* gene was detected in 3D4/21 cells infected by H1N1 and H3N2, but no expression of *NP* gene was detected in non-infected (NC) cells. Similarly, NP protein could be detected in 3D4/21 cells infected by H1N1 and H3N2, but could not be detected in NC cells (Figure 1I).

In summary, H1N1 and H3N2 could replicate and proliferate well in 3D4/21 cells after cells were infected by H1N1 and H3N2 with 100-fold TCID_50_ for 48 h, which caused greater changes in the expression levels of virus-related genes and stronger changes in the secretion levels of cytokines. Therefore, the 3D4/21 cells infected by H1N1 and H3N2 with 100-fold TCID_50_ for 48 h in this study was used to construct virus infection cell models.

### 2.2. Whole Transcriptome Sequencing Analysis of 3D4/21 Cells Infected by H1N1 and H3N2

The RNA-Seq sequencing data (Reference genome: https://www.ncbi.nlm.nih.gov/genome/?term=pig, accessed on 12 December 2019) was compared and analyzed by Hisat2 software (2.1.0) (http://ccb.jhu.edu/software/hisat2, accessed on 8 June 2017) [13]. The overall statistics of lncRNA sequencing data were shown in Appendix A, and the overall statistics of miRNA sequencing data were shown in Appendix A. DESeq2 R software (1.12.0) was used to analyze the significance of differentially expressed mRNAs, lncRNAs, circRNAs and miRNAs [14]. As shown in Appendix A, compared with NC group, there are 119 mRNAs, 57 lncRNAs, 22 circRNAs and 5 miRNAs that were common differentially expressed both in H1N1 group and H3N2 group. Some common differential mRNAs, lncRNAs, circRNAs and miRNAs were listed in Table 1.

GO enrichment (Appendix A) and KEGG enrichment analysis (Appendix A) were performed on all differentially expressed mRNAs, lncRNAs, circRNAs and miRNAs. The results showed that the differential mRNAs were mainly enriched in GO classifications including metabolic processes, receptor binding and extracellular areas, as well as pathways including TNF signaling pathway, NOD-like receptor signaling pathway, and influenza A. The target genes of differential lncRNAs were mainly enriched in GO classifications including multicellular biological processes, signal receptor activity and signal transduction activity, as well as pathways including mRNA monitoring pathways, ribosomal and NF-κB signaling pathways. The source genes of differential circRNAs were mainly enriched in GO classifications including the metabolic processes of organic ring compounds, intracellular parts and cellular metabolic processes, as well as pathways including insulin signaling pathway, mTOR signaling pathway, and mRNA monitoring pathway. The target genes of differential miRNAs were mainly enriched in GO classifications including protein binding, cytoplasm and intracellular organelles, as well as pathways including NF-κB signaling pathways, glycosphingolipid biosynthesis-globulin series and viral carcinogenic pathways. Finally, qPCR analysis was performed on some of the differential mRNAs, miRNAs, lncRNAs and circRNAs, and it was found that (Figure 2) the expression trends of differential mRNAs, miRNAs, lncRNAs and circRNAs were consistent with the sequencing results.

### 2.3. CeRNA Network in 3D4/21 Cells Infected by H1N1 and H3N2

Combined the results of whole transcription sequencing and bioinformatics analysis (Table 2, Table 3 and Table 4), possible ceRNA regulatory networks were constructed in this study based on the common differential miRNAs, lncRNAs, mRNAs and circRNAs (Figure 3A). Among them, miR-10391, novel_circ_0004733, TCONS_00166432 and *MAN2A1* gene are common differentially expressed in the H1N1 and H3N2 groups. Therefore, two potential ceRNA networks (novel_circ_0004733-miR-10391-MAN2A1 and TCONS_00166432-miR-10391-MAN2A1) were selected in this study for further verification.

The expression levels of TCONS_00166432 and novel_circ_0004733 in cytoplasmic RNA and nuclear RNA were detected by qPCR. The results showed that TCONS_00166432 expressed about 49.2% in the nucleus and 50.8% in the cytoplasm; novel_circ_0004733 expressed about 12.9% in the nucleus and 87.1% in the cytoplasm (Figure 3B). As shown in Figure 3C, the seed sequences of miR-10391 completely matched the 3′ UTR sequences of target gene *MAN2A1*. In addition, miR-10391 had 7 base matched with novel_circ_0004733 and TCONS_00166432, respectively. This result indicated that miR-10391 had targeted binding sites with *MAN2A1*, novel_circ_0004733 and lncRNA TCONS_00166432.

The overexpression vectors and interference vectors of lnc_TCONS_00166432 and novel_circ_0004733 were constructed, respectively (Appendix A). 3D4/21 cells were transfected with miR-10391-inhibitor, miR-10391-mimics, TCONS_00166432-pcDNA3.1, TCONS_00166432-sh1, TCONS_00166432-sh2, novel_circ_0004733-pcDNA3.1, novel_circ_0004733-sh1 and novel_circ_0004733-sh2, respectively (Figure 3D–F). The relative expression levels of miR-10391, TCONS_00166432 and novel_circ_0004733 after transfection were detected by qPCR. As shown in Figure 3G, miR-10391-mimics could significantly up-regulate the expression level of miR-10391 (*p* < 0.01), miR-10391-inhibitor could significantly down-regulate the expression level of miR-10391 (*p <* 0.05). Similarly, TCONS_00166432-pcDNA3.1 could significantly up-regulate the expression level of TCONS_00166432 (*p <* 0.01) and TCONS_00166432-sh2 could significantly down-regulate the expression level of miR-10391 (*p <* 0.05). Novel_circ_0004733-pcDNA3.1 could significantly up-regulate the expression level of TCONS_00166432 (*p <* 0.01) and novel_circ_0004733-sh1 could significantly down-regulate the expression level of miR-10391 (*p <* 0.05). Therefore, miR-10391-inhibitor, miR-10391-mimics, TCONS_00166432-pcDNA3.1, TCONS_00166432-sh2, novel_circ_0004733-pcDNA3.1 and novel_circ_0004733-sh1 were chosen for further research.

The results of dual luciferase activity detection revealed the targeting relationship of miR-10391 with *MAN2A1* gene, novel_circ_0004733 and TCONS_00166432 (Figure 3H). MiR-10391-mimics could down-regulate the activity of MAN2A1-wt significantly (*p <* 0.01), while miR-10391-inhibitor up-regulated the activity of MAN2A1-wt significantly (*p <* 0.01). MiR-10391-mimics down-regulated the activity of circ-wt significantly (*p <* 0.05), while miR-10391-inhibitor up-regulated the activity of circ-wt significantly (*p <* 0.01). MiR-10391-mimics down-regulated the activity of lnc-wt significantly (*p <* 0.05), while miR-10391-inhibitor up-regulated the activity of lnc-wt significantly (*p <* 0.01). As shown in Figure 3I, the RIP experiment further verified the targeting relationship between MAN2A1 and miR-10391 as well as TCONS_00166432.

### 2.4. The Regulatory Role of miR-10391, TCONS_00166432 and novel_circ_0004733 in the Process of H1N1 and H3N2 Infecting 3D4/21 Cells

The effects of miR-10391, TCONS_00166432 and novel_circ_0004733 on the expression of viral genes and host genes were detected by qPCR, and the secretion levels of cytokines were detected by ELISA. As shown in Figure 4A,B, the expression level of miR-10391 affected the expression of *NLRP3* gene in host cells after infected 3D4/21 cells were infected by H1N1 and H3N2 (*p <* 0.01). However, it had no significant effect on the expression levels of viral genes *M* and *NP* (*p >* 0.05). In addition, the expression level of miR-10391 affected the secretion levels of IFN-α, IFN-β and IFN-γ in the cell culture supernatant after 3D4/21 cells were infected by H1N1 (*p <* 0.05) (Figure 4G). The expression level of miR-10391 affected the secretion levels of IFN-α and IFN-γ in the cell culture supernatant after 3D4/21 cells were infected by H3N2 (*p <* 0.05) (Figure 4H).

As shown in Figure 4C,D, the expression level of TCONS_00166432 had no significant effect on the host cell genes (*RIG-I*, *TLR7* and *NLRP3*) after H1N1 infected 3D4/21 cells (*p >* 0.05). However, it affected the expression of host cell genes *TLR7* and *NLRP3* after 3D4/21 cells were infected by H3N2 (*p <* 0.05). In addition, the expression level of TCONS_00166432 could significantly affect the expression of viral gene *NP* after H3N2 infected 3D4/21 cells (*p <* 0.05). As shown in Figure 4I,J, the expression level of TCONS_00166432 affected the secretion levels of IFN-α and IFN-γ in the cell culture supernatant after 3D4/21 cells were infected by H1N1 (*p <* 0.05). The expression level of TCONS_00166432 affected the secretion levels of IFN-β and IFN-γ in the cell culture supernatant after 3D4/21 cells were infected by H3N2 (*p <* 0.05).

As shown in Figure 4E, the expression level of novel_circ_0004733 significantly affected the expression of host genes (*RIG-I*, *TLR7* and *NLRP3*) and the expression of viral gene *NP* after 3D4/21 cells were infected by H1N1 (*p <* 0.05 or *p <* 0.01). As shown in Figure 4F, the expression level of novel_circ_0004733 significantly affected the expression of host cell genes (*RIG-I*, *TLR7* and *NLRP3*) (*p <* 0.01) and the expression of viral genes *M* and *NP* (*p <* 0.05) after 3D4/21 cells were infected by H3N2. As shown in Figure 4K,L, the expression level of novel_circ_0004733 significantly affected the secretion level of IFN-γ in the cell culture supernatant after 3D4/21 cells were infected by H1N1 and H3N2 (*p <* 0.01).

### 2.5. The Regulatory Role of MAN2A1 Gene in the Process of H1N1 and H3N2 Infecting 3D4/21 Cells

In this study, the shRNA interference vectors MAN2A1-sh1, MAN2A1-sh2 and MAN2A1-sh3 of the *MAN2A1* gene were constructed (Appendix A) and transfected into 3D4/21 cells (Figure 5A). The qPCR results showed that the relative expression levels of *MAN2A1* were down-regulated significantly after 3D4/21 cells were transfected by three interference vectors (*p <* 0.01), the relative expression levels were 0.37, 0.28 and 0.22, respectively (Figure 5B). The Western blot results showed that the protein levels of MAN2A1-sh2 and MAN2A1-sh3 also decreased significantly. Therefore, the MAN2A1-sh3 vector with the highest interference efficiency was chosen for further study.

As shown in Figure 5C,E, the interference of *MAN2A1* gene had no significant effect on host cell genes (*RIG-I*, *TLR7* and *NLRP3*) (*p* > 0.05). However, the interference of *MAN2A1* gene down-regulated the expression levels of H1N1 and H3N2 viral genes *M* and *NP* significantly (*p <* 0.05 or *p <* 0.01). As shown in Figure 5D,F, *MAN2A1* gene interference down-regulated the secretion levels of IFN-α and IFN-γ in 3D4/21 cells infected by H1N1 significantly (*p <* 0.05), and down-regulated the secretion level of IFN-α in 3D4/21 cells infected by H3N2 significantly (*p <* 0.05).

In this study, three core promoters were predicted in the sequence of about 1000 bp upstream of the transcription start site (TSS) of the *MAN2A1* gene. Three core promoters were located in 742–792 bp, 584 bp-634 bp and 126 bp-176 bp upstream of the TSS, and two SNP sites (G/T mutation and A/T mutation) were found in the second core promoter region (Figure 5G). Then primers were designed to amplify the promoters with different fragments and the firefly luciferase vectors were constructed (Appendix A). The results of dual luciferase activity assay showed that the activity of promoter fragment was down-regulated significantly at −282-81 (*p <* 0.05), and down-regulated at −78–81 significantly (*p <* 0.01) (Figure 5H). It indicated that there might be transcriptional regulatory elements in the two fragments between 78–282 and 282–679. Interestingly, the predicted core promoter region (584 bp-634 bp and 126 bp-176 bp upstream of the TSS) were also located in these two fragments. Our results also verified the reliability of core promoter prediction.

The wild-type (WT) and mutant (G/T and A/T) oligos referring to the two SNP sites in the second core promoter of *MAN2A1* gene were synthesized and ligated to the pGL3-basic vector to construct firefly luciferase expression vectors (Appendix A). The effect of the mutation sites on the transcription activity of the *MAN2A1* gene was detected by dual luciferase activity assay, and the results showed that the A/T mutation (587 bp upstream of the TSS) increased the transcription activity significantly (*p <* 0.05) (Figure 5I). Further prediction through the website showed that the SNP sequence in the core promoter region of the *MAN2A1* gene might bind to three transcription factors: MRF4, PEA3 and Sp1 (Figure 5J).

### 2.6. CeRNA Regulation Mechanism in 3D4/21 Cells Infected by H1N1 and H3N2

In this study, we illustrated that miR-10391 could inhibit the expression of *MAN2A1* gene by binding the 3′UTR of *MAN2A1* gene, while novel_circ_0004733 and TCONS_00166432 could act as the sponge of miR-10391 to reduce the effect of miR-10391 on *MAN2A1* gene by binding the sequences of miR-10391. In conclusion, TCONS_00166432 and novel_circ_0004733 acted as miRNA sponges to regulate the expression of miR-10391, which targeted the *MAN2A1* gene and regulated the replication and proliferation of SIV in 3D4/21 cells (Figure 6).

## 3. Discussion

The SIV genome consists of 8 RNA fragments. They are PB1, PB2, and PA encoding polymerase, NP encoding nucleoprotein, HA encoding hemagglutinin, NA encoding neuraminidase, M (M1 and M2) encoding matrix protein, and NS (NS1 and NS2) encoding non-structural proteins [15]. Among them, the nucleoprotein NP is a monomer phosphorylated polypeptide, the main component of the viral nucleocapsid, and the most important diagnostic protein for influenza viruses. Matrix protein M1 can maintain the shape of the virus and can also be used as a basis for influenza virus typing. Matrix protein M2 is a trans-matrix protein on the host cell membrane, which plays a role of proton channel and enables viral ribonucleoprotein (RNP) to enter the cytoplasm. Therefore, NP and M are used as the detection markers for virus replication and proliferation in 3D4/21 cells. Moreover, we found that H1N1 and H3N2 could replicate and proliferate well in 3D4/21 cells after cells were infected by H1N1 and H3N2 with 100-fold TCID_50_ for 48 h.

Viral RNA in infected cells can be recognized by the pattern recognition receptor PRR, and can be divided into Toll-like receptor (TLR), RIG-I-like receptor (RLR) and Nueleotide oligomerization domain-like receptor (NLR). Among them, the *TLR3* gene in the Toll-like receptor, the *RIG-1* gene in the RIG-I-like receptor and the *NLRP3* gene in the NOD receptor have been shown to recognize influenza viruses [16,17,18]. Therefore, *RIG-I*, *NLRP3* and *TLR7* were used as the detection markers of immune response in 3D4/21 cells infected by virus. Moreover, we found that the expression of these genes in 3D4/21 cells infected by H1N1 and H3N2 with 100-fold TCID_50_ for 48 h increased significantly. Type I interferons (IFN-α and IFN-β) mainly play an antiviral role, and type II interferons (mainly IFN-γ) mainly play an immunomodulatory role [19]. These three cytokines are commonly used as the detection markers of host immune response caused by influenza virus infection. Moreover, we found that strong immune response appeared in 3D4/21 cells infected by H1N1 and H3N2 with 100-fold TCID_50_ for 48 h. In summary, the 3D4/21 cells model constructed in this study could be used to further analyze the molecular mechanism of SIV infecting 3D4/21 cells.

Whole transcriptome sequencing provides a more comprehensive research method, which is of great significance for analyzing the pig growth and development and the molecular regulation mechanism of pig disease occurrence [20]. We identified and analyzed the differential expression of mRNAs, lncRNAs, miRNAs and circRNAs after H1N1 and H3N2 infected 3D4/21 cells, and focused on RNAs that were common differentially expressed both in the H1N1 and H3N2 groups compared with NC group. In this study, a total of 119 common differential mRNAs were screened out in the H1N1 and H3N2 groups. Among them, MAN2A1 (mannosidase alpha class 2A member 1) is a Golgi enzyme that converts high mannose into a complex structure of N-glycans to mature and glycosylate membrane proteins, which plays an important biological function in tumor and other immune processes [21]. In this study, 57 co-differentially expressed lncRNAs, 22 co-differentially expressed circRNAs and 5 co-differentially expressed miRNAs were screened in the H1N1 and H3N2 groups. These lncRNAs, circRNAs and miRNAs have not been reported to be involved in the regulation of SIV infection. Therefore, this study revealed for the first time the expression profiles of mRNAs, lncRNAs, circRNAs and miRNAs after H1N1 and H3N2 infected 3D4/21 cells at the transcriptome level.

Studies have shown that lncRNA expressed in the nucleus is mainly involved in regulating transcription, chromatin and variable splicing; lncRNA expressed in the cytoplasm is mainly involved in regulating ceRNA, mRNA stability and translation [22,23]. In this study, TCONS_00166432 was expressed both in the nucleus and cytoplasm, suggesting that TCONS_00166432 may not only play a regulatory function through its neighboring gene (*MAN2A1*), but also may affect the expression of the target gene *MAN2A1* through the ceRNA network (TCONS_00166432-miR10391-MAN2A1). Similarly, circRNA is distributed in both the cytoplasm and the nucleus. Studies have found that exon circRNA in the cytoplasm can be used as the sponge molecule of miRNA, mainly regulating transcription and post-transcriptional modification [24,25]. In this study, novel_circ_0004733 was mainly expressed in the cytoplasm, suggesting that novel_circ_0004733 may mainly regulate the expression of target genes through the ceRNA network (novel_circ_0004733-miR10391-MAN2A1). In addition, the results of dual luciferase activity assay revealed that miR-10391 could target *MAN2A1* gene, novel_circ_0004733 and TCONS_00166432. Finally, the results of the RIP experiment further illustrated that miR-10391 could target the *MAN2A1* gene, and TCONS_00166432 could target the *MAN2A1* gene. Therefore, we reported for the first time the ceRNA network involved in the regulation of the SIV infecting 3D4/21 cells, and provided new insights for revealing the molecular mechanism of 3D4/21 cells resisting SIV infection.

In this study, it was found that the expression level of miR-10391 affected the expression of *NLRP3* gene in host cells after H1N1 and H3N2 infected 3D4/21 cells significantly, but it had no significant effect on the expression levels of viral genes *M* and *NP*. In addition, the expression level of miR-10391 affected the secretion levels of IFN-α and IFN-γ in the cell culture supernatant after H1N1 and H3N2 infected 3D4/21 cells. Guo et al. [26] revealed that the up-regulation of miR-181 could directly inhibit PRRSV replication and had an impact on the control of viral infections. However, the results of this study suggested that the expression of miR-10391 might not directly regulate the replication and proliferation of influenza virus, but may participate in the intracellular immune response after SIV infection. Up-regulation of miR-10391 suppressed the strength of the immune response to a certain extent, thereby reducing the damage of the virus to the host cell. Núñez-Hernández et al. [27] analyzed the miRNAs in the tonsils and mediastinal lymph nodes (MLN) of pigs before and after PCV2 infection, and found that some differentially expressed miRNAs may be involved in pathways related to the immune system and processes related to the pathogenesis of PCV2. We speculated that miR-10391 also affected the regulation process of SIV infecting host cells by participating in the immune system-related pathways.

In this study, it was found that the expression level of TCONS_00166432 could affect the expression of viral gene *NP* in 3D4/21 cells infected by H3N2 significantly. The results suggested that the expression of TCONS_00166432 might directly regulate the replication level of virus gene *NP*, and the low expression of TCONS_00166432 might be beneficial to inhibit virus replication. In addition, the expression level of TCONS_00166432 affected the secretion level of IFN-γ in the cell culture supernatant after H1N1 and H3N2 infected 3D4/21 cells. The results suggested that the expression of TCONS_00166432 might be involved in regulating the immune response of host cells to a certain extent, and the down-regulation of TCONS_00166432 might help 3D4/21 cells to resist SIV infection. Zhang et al. [28] analyzed the long non-coding RNA (lncRNA) in porcine alveolar macrophages (PAM) 12 and 24 h after cells were infected by PRRSV. Wu et al. [29] identified some of the differentially expressed lncRNAs in PAM infected by PRRSV were related to interferon-induced genes, and these lncRNAs may play an important role in the host’s innate immune response to PRRSV infection. In this study, we speculated that TCONS_00166432 could regulate the infection of SIV through a similar action pathway.

In this study, it was found that the expression level of novel_circ_0004733 significantly affected the expression of host cell genes (*RIG-I*, *TLR7* and *NLRP3*) and the expression of viral genes *M* and *NP* after H1N1 and H3N2 infected 3D4/21 cells. The results suggested that the expression of novel_circ_0004733 may directly regulate the replication level of SIV *M* gene and *NP* gene, and up-regulation of novel_circ_0004733 may be beneficial to virus replication and proliferation. In addition, the expression level of novel_circ_0004733 significantly affected the secretion level of IFN-γ in the cell culture supernatant after H1N1 and H3N2 infected 3D4/21 cells. The results suggested that the expression of novel_circ_0004733 is involved in regulating the immune response of host cells to a certain extent, and the down-regulation of novel_circ_0004733 might help 3D4/21 cells to resist SIV infection.

MAN2A1 is an enzyme encoded in the maturation of N-glycans. As a key immunomodulatory gene, the absence of MAN2A1 in cancer cells increases their sensitivity to T cell-mediated killing [30]. In this study, it was found that the interference of *MAN2A1* directly significantly down-regulated the expression levels of virus genes *M* and *NP* while it had no significant effect on the expression level of host genes (*RIG-I*, *TLR7* and *NLRP3*) in the cells after SIV infected 3D4/21 cells. The results suggested that the expression of *MAN2A1* may mainly regulate the infection of SIV by regulating the replication and proliferation of SIV instead of innate immune response. In addition, ELISA results showed that the interference of *MAN2A1* down-regulated the secretion level of IFN-α in cells after virus infection significantly, suggesting that the expression of *MAN2A1* affected the cellular immune response to a certain extent. Finally, we found that the A/T mutation of the SNP in *MAN2A1* gene promoter region could significantly affect transcriptional activity and could be used as a potential molecular marker for disease resistance breeding.

In summary, porcine alveolar macrophage cell line (3D4/21) models infected by H1N1 and H3N2 were established in this study. Secondly, the expression profiles of mRNAs, lncRNAs, circRNAs and miRNAs after H1N1 and H3N2 infected 3D4/21 were illustrated for the first time. Thirdly, the ceRNA regulatory network for 3D4/21 cells to resist SIV infection was constructed and verified. Finally, the regulation mechanism of differential miRNA, lncRNA and circRNA and target gene (*MAN2A1*) on 3D4/21 cells infected by H1N1 and H3N2 was illustrated, which provided new insights for the analysis of the molecular regulation mechanism of SIV infecting host cells. It was initially revealed that the SNP in the core promoter region of the *MAN2A1* gene could significantly affect the expression of the *MAN2A1* gene, which can be used as a potential molecular marker against SIV for verification and research.

## 4. Materials and Methods 

### 4.1. Primer Design and Synthesis

All primer sequences in this study were designed through the online website (https://primer3.ut.ee/, Primer3web version 4.1.0, accessed on 8 June 2020), using *GAPDH* as the internal reference gene of mRNA, lncRNA and circRNA, and *U6* (Tiangen Biochemical Technology Co., Ltd., Beijing, China) as the internal reference gene of miRNA. All of the primers information were shown in Appendix A and synthesized by Sangon Biotech Co., Ltd. (Shanghai, China).

### 4.2. Proliferation and TCID_50_ Determination of H1N1 and H3N2 

Virus strains “A/swine/Liaoning/32/2006 (H1N1) and A/swine/Heilongjiang/10/2007 (H3N2) were gifts from Professor Guoqiang Zhu, College of Veterinary Medicine, Yangzhou University. The H1N1 and H3N2 virus fluids were added to the subcultured Madin-Daby canine kidney cells (MDCK), respectively. Cells were incubated for 2 h in a 37 °C incubator and then virus fluids were aspirated, replaced with fresh DMEM medium. Cells were continued to be cultured until obvious cytopathic alterations were observed under the microscope. Then cells were repeatedly frozen and thawed three times to release the virus particles. The supernatant was collected by centrifugation and stored at −80 °C for later use. In addition, MDCK cells were subculture into two 96-well plates, H1N1 and H3N2 virus fluids were added to plates, respectively. Use medium DMEM to make 10-fold dilutions of the virus fluids, count the diseased cells with different virus gradients. Reed-Muench method [12] was used to calculate the 50% tissue culture infective dose (TCID_50_) of the virus.

### 4.3. 3D4/21 Cells Are Infected by H1N1 and H3N2

3D4/21 cells were cultured with 1640 complete medium containing 10% fetal bovine serum in incubator under the condition of 37 °C and 5% CO_2_. Then cells were subcultured into two 24-well plates, and H1N1 and H3N2 virus fluids were added to plates, respectively. Three time gradients (24 h, 48 h and 72 h) and three dose gradients (1-fold TCID_50_, 10-fold TCID_50_ and 100-fold TCID_50_) were set in this study. The cells were incubated in a 37 °C incubator for 2 h and then virus fluids were aspirated, replaced with fresh 1640 complete medium. The total RNA of treated cells was extracted at 24 h, 48 h, and 72 h according to the instructions of Trizol reagent (Vazyme Biotech Co., Ltd, Nanjing, China.). At the same time, the cell culture supernatant was collected for ELISA determination.

### 4.4. cDNA Synthesis and qPCR Detection

This research involved the expression levels detection of mRNA, miRNA, lncRNA and circRNA. CDNA was synthesized according to instructions (HiScript 1st Strand cDNA Synthesis Kit, Vazyme Biotech Co., Ltd., Nanjing, China; miRcute Plus miRNA First-Strand cDNA Kit, Tiangen, China; lnRcute lncRNA First-Strand cDNA Kit, Tiangen, China). QPCR detection was conduced following the instructions, respectively (Taq Pro Universal SYBR qPCR Master Mix, Vazyme Biotech Co.,Ltd, Nanjing, China; miRcute Plus miRNA qPCR Kit, Tiangen, China; lnRcute lncRNA qPCR Kit, Tiangen, China; circRNA fluorescence quantification kit, Geneseed, China). Finally, the relative expression level was calculated by the 2^−ΔΔCt^ method, and each treatment was conducted with three replicates [31].

### 4.5. Western Blotting

3D4/21 cells were subcultured to a 6 cm cell culture dish and were infected by 100-fold TCID_50_ H1N1 and H3N2, respectively. Control cells without virus infection were set at the same time. After 48 h, the cells were carefully rinsed with PBS for three times, 300 μL of protein lysis buffer was added and the cells were placed on ice for 30 min. Cells were scraped with cell scrapers and the supernatant was collected by centrifugation at 4 °C for 20 min. Then the protein concentration was measured with the BCA kit (Cowin Bio., Taizhou, China). 5× protein buffer was added to samples and then boiled at 98 °C for 10 min. SDS-PAGE of the protein samples (10 µL) was performed at 120 V for 90 min in a 10% gel. Protein sample was transferred to a PVDF membrane and immunoblotted with the relevant antibody. Blocking solution and antibodies (NP, MAN2A1, GAPDH; Abcam, UK) were added at approximately 0.1 mL/cm^2^. Second antibody (IgG, Cowin Bio., Taizhou, China) was added after samples were washed by PBST for three times. ECL luminescence reagent was added to PVDF membrane and exposed on the chemiluminescence imager.

### 4.6. ELISA Detection

The cell culture supernatant processed in the previous step was collected and the the cytokines (IFN-α, IFN-β and IFN-γ) secretion levels were detected by ELISA kits (Nanjing Jiancheng Bioengineering Institute, Nanjing, China) according to the instruction.

### 4.7. Whole Transcriptome Sequencing Analysis

#### 4.7.1. Samples Preparation

3.D4/21 cells were subcultured to 12 T25 cell culture flasks and the virus infection experiment was carried out after the number of cells reached 10^7^. Cells were infected by H1N1 and H3N2 with 100-fold TCID_50_ dose, respectively. There were 3 treatment groups (H1N1 infection group, H3N2 infection group and negative control (NC) group) and 4 replicates for each treatment. 48 h later, it was observed under the microscope that the cells in the virus-infected group began to develop lesions, while the shape of the NC group was normal. Cells pellets were collected in 12 cryopreservation tubes after centrifugation. 1 mL of trizol was added to each tube and mix well by pipetting, tubes were labeled with H1N1_1, H1N1_2, H1N1_3, H1N1_4, H3N2_1, H3N2_2, H3N2_3, H3N2_4, NC_1, NC_2, NC_3, and NC_4, respectively. Beijing Novogene Technology Co., Ltd. (Beijing, China) was entrusted to perform whole transcriptome sequencing analysis.

#### 4.7.2. RNA Quality Control, Library Construction and Sequencing

The quality of total cell RNA was detected by agarose gel electrophoresis, Nanodrop, Qubit and Agilent 2100. In this study, on the one hand, ribosomal RNA was removed to construct a chain-specific library [32], including lncRNA, mRNA and circRNA. After the samples were subjected to quality control, Illumina PE150 was used for sequencing. On the other hand, the small RNA library was constructed with the Small RNA Sample Pre Kit. After the samples were subjected to quality control, HiSeq/MiSeq was used for sequencing. After obtaining the raw data, the quality of the sequencing data was evaluated by calculating the error rate, data volume and comparison rate. Then quality control, comparison, splicing, screening, quantification, significant difference analysis and functional enrichment as well as analysis of variable splicing and mutation sites related to transcript structural variation was performed int this study. Finally, statistical methods were used to compare gene expression differences between different treatment groups (H1N1 infection group, H3N2 infection group and NC group), to find out the relevant differential genes and analyze their biological significance. 

### 4.8. ceRNA Mechanism Research

#### 4.8.1. ceRNA Network Prediction

Based on 5 co-differentially expressed miRNAs in the H1N1 and H3N2 infection groups compared to NC group, candidate lncRNA-miRNA-mRNA networks and circRNA-miRNA-mRNA networks were constructed according to the predicted targeted binding mRNAs, lncRNAs and circRNAs.

#### 4.8.2. 3D4/21 Cytoplasmic and Nuclear RNA Extraction and Expression Level Analysis

In order to determine the expression location of lncRNA (TCONS_00166432) and circRNA (novel_circ_0004733) in 3D4/21 cells, cytoplasmic and nuclear RNA was separated by cytoplasmic and nuclear RNA extraction reagents (Amyjet, Wuhan, China) in this study. The nuclear reference gene *U6* and the cytoplasmic reference gene *GAPDH* were used for quality control and the expression levels of TCONS_00166432 and novel_circ_0004733 in total cell RNA, cytoplasmic RNA and nuclear RNA were detected by qPCR.

#### 4.8.3. Design and Synthesis of miR-10391 Mimics and Inhibitor

According to the mature sequence of pig miR-10391 provided by miRBase database (http://www.mirbase.org/cgi-bin/mirna_entry.pl?acc=MI0033405, accessed on 2 May 2020), mimics sequence (5’-3’) was designed as F: AAGGAAGGAGACUAACUCCGCC; R: CGGAGUUAGUCUCCUUCCUUUU. Mimics sequence (5’-3’) was designed as: GGCGGAGUUAGUCUCCUUCCUU. Both miR-10391 mimics and miR-10391 inhibitor sequences were synthesized by GenePharma Biotech Co., Ltd. (Suzhou, China), and the corresponding control siRNAs (mimics-NC and inhibitor-NC) were provided.

#### 4.8.4. Overexpression/Interference Vector Construction of TCONS_00166432 and novel_circ_0004733

The whole genome sequences of TCONS_00166432 and novel_circ_0004733 were synthesized by GenePharma Biotech Co., Ltd., with adding restriction sites (*Xho*I and *Xba*I) and then the sequences were ligated to the pcDNA3.1(+) vector. The recombinant vectors were extracted and purified by using a EndoFree Mini Plasmid Kit (Tiangen, Beijing, China), and they were named as TCONS_00166432-pcDNA3.1 and novel_circ_0004733-pcDNA3.1.

According to the TCONS_00166432 and novel_circ_0004733 sequences provided by the whole transcriptome sequencing results, Invitrogen RNAi Designer software was used to design and synthesize two short hairpin RNA (shRNA) interference sequences for TCONS_00166432 and novel_circ_0004733, respectively (Appendix A). The sequences were ligated to pGPU6/GFP/Neo vector, respectively. The recombinant vectors were extracted and purified by using a EndoFree Mini Plasmid Kit (Tiangen, Beijing, China), and they were named as TCONS_00166432-sh1, TCONS_00166432-sh2, novel_circ_0004733-sh1 and novel_circ_0004733-sh2.

#### 4.8.5. Recombinant Vectors Transfected Cells and qPCR Detection

3D4/21 cells were subcultured into a 12-well plate, and 1 mL of 1640 complete medium was added to each well, with cell density about 50%. The ssc-miR-10391 mimics, mimic-NC, ssc-miR-10391 inhibitor, inhibitor-NC, TCONS_00166432-pcDNA3.1, novel_circ_0004733-pcDNA3.1, pcDNA3.1-GFPTCONS_00166432-sh1, lnc-TCONS_00166432-sh2, novel_circ_0004733-sh1, novel_circ_0004733-sh2 and pGPU6/GFP/Neo-shNC was transfected to 3D4/21 cells according to the instructions of jet PRIME transfection reagent (Polyplus Transfection, Illkirch-Graffenstaden, France).

For siRNA (miR-10391 mimics and miR-10391 inhibitor), green fluorescence was observed after transfection for 24–48 h. Total cell RNA was extracted and cDNA was synthesized. Then the relative expression level of miR-10391 in the cells of each treatment group was detected by qPCR.

For shRNA and pcDNA-3.1 vectors, G418 (600 mg/mL) was added after transfection for 24 h. After cells stably expressed green fluorescent protein, total cell RNA was extracted and cDNA was synthesized. Then the relative expression levels of TCONS_00166432 and novel_circ_0004733 in the cells of each treatment group were detected by qPCR.

#### 4.8.6. Construction of Firefly Luciferase Vectors for *MAN2A1* Gene 3′UTR, TCONS_00166432 and novel_circ_0004733

In order to further determine the targeting relationship between miR-10391, *MAN2A1* gene, TCONS_00166432 and novel_circ_0004733, wild-type and mutant oligos were designed (Appendix A). The fragments were ligated to the pMIR-Report Luciferase vector to construct the corresponding firefly luciferase vectors, respectively. The recombinant vectors were extracted with a EndoFree Mini Plasmid Kit, and the recombinant plasmids were named as MAN2A1-wt, MAN2A1-mut, lncRNA-wt, lncRNA-mut, circRNA-wt, and circRNA-mut.

#### 4.8.7. Cell Transfection and Dual Luciferase Activity Detection

293T cells were cultured with DMEM medium containing 10% fetal bovine serum in a 24-well plate. When cell density reached about 70%, the recombinant vectors and pRL-TK vector were co-transfected into miR10391-mimics cells and miR10391-inhibitor cells, and the cells were collected after 48 h. The Dual Luciferase Reporter Assay Kit (Vazyme Biotech Co., Ltd., Nanjing, China) was used to detect the effect of miRNA on luciferase activity. The ratio of firefly luciferase Ff activity/renin luciferase Rn activity was regarded as luciferase activity.

#### 4.8.8. RNA Immunoprecipitation (RIP)

In order to further determine the targeting relationship between *MAN2A1* and TCONS_00166432 as well as miR-10391, RNA and protein in the cells were extracted and verified by RNA immunoprecipitation (RIP) according to the instruction of RNA Immunoprecipitation Kit (Geneseed, Guangzhou, China). The RNA obtained from the RIP was subjected to reverse transcription and tested by qPCR to detect the expression levels of *MAN2A1*, TCONS_00166432 and miRNA. The protein obtained from the RIP test was detected by Western blotting with *MAN2A1* protein.

### 4.9. The regulatory Role of miR-10391, TCONS_00166432 and novel_circ_0004733 in the Process of H1N1 and H3N2 Infecting 3D4/21 Cells

H1N1 and H3N2 were used to infect 3D4/21 cells of different treatment groups (miR-10391-mimics group, miR-10391-inhibitor group, TCONS_00166432 overexpression group, TCONS_00166432 interference group, novel_circ_0004733 overexpression group, novel_circ_0004733 interference group and negative control group). After 48 h, the cell total RNA was collected for qPCR analysis, and the cell culture supernatant was collected for ELISA analysis.

### 4.10. The Regulatory Role of MAN2A1 Gene in the Process of H1N1 and H3N2 Infecting 3D4/21 Cells

According to the CDS region sequence of the pig *MAN2A1* gene (NCBI accession number: XM003123823.6), three shRNA sequences targeting the *MAN2A1* gene were designed using Invitrogen RNAi Designer software (Appendix A). Recombinant vectors (MAN2A1-sh1, MAN2A1-sh2, MAN2A1-sh3, and shRNA-NC) were constructed and transfected into 3D4/21 cells, and G418 (600 mg/mL) was added to cells after the cells were transfected for 24 h. When the cells stably expressed green fluorescent protein, the cell total RNA was extracted and cDNA was synthesized. Then the expression of *MAN2A1* gene was detected by qPCR and the expression of *MAN2A1* protein was detected by Western blot.

H1N1 and H3N2 were used to infect 3D4/21 cells in different treatment groups (MAN2A1-shRNA group, and NC group). After cells were infected for 48 h, the cell total RNA was collected for qPCR analysis, and the cell culture supernatant was collected for ELISA analysis.

### 4.11. Regulatory Mechanism of MAN2A1 Gene Expression

The 1000 bp sequence upstream of the transcription start site (TSS) of the *MAN2A1* gene was selected, and primers (Appendix A) were designed to amplify the different fragments of promoter. The PCR products were recovered and purified, and then double digestion (*Hind*III and *Nco* I) was performed for PCR products and pGL3-basic vector, respectively. After ligation and transformation, recombinant vectors were constructed and the plasmids were extracted. 293T cells were cultured with DMEM medium containing 10% fetal bovine serum in a 12-well plate. When the cell density reached 80%, the recombinant luciferase vectors and Renilla Luciferase vector pRL-TK were co-transfected into 293T cells. After 48 h, the cells were collected and the dual luciferase activity assay was performed according to the instructions.

The core promoter region of *MAN2A1* gene was predicted by BDGP software (http://www.fruitfly.org/seq_tools/promoter.html, NNPP version 2.2, accessed on 8 June 2020). The SNP information in the core promoter region was derived from the Ensembl database, and wild-type and mutant oligo in the core promoter region of the *MAN2A1* gene were synthesized (Appendix A). The fragments were recombined into the pGL3-basic vector. Finally, the effect of different mutation sites on the promoter activity was analyzed by Dual Luciferase Reporter Assay Kit.

### 4.12. Statistical Analysis

The 2^−ΔΔCt^ method was used to analyze the relative quantitative results [31], and the internal reference gene was used to homogenize the expression level of the target gene. The student’s t test of SPSS 17.0 software was used to compare the relative expression levels of miRNA, mRNA, lncRNA and circRNA in the transfected cells and the control cells. General Linear Model (GLM) of SPSS 17.0 software was used to analyze and compare related genes expression levels and cytokine levels at different time and with different doses of cells infected by H1N1 and H3N2. 

## Figures and Tables

**Figure 1 ijms-23-01875-f001:**
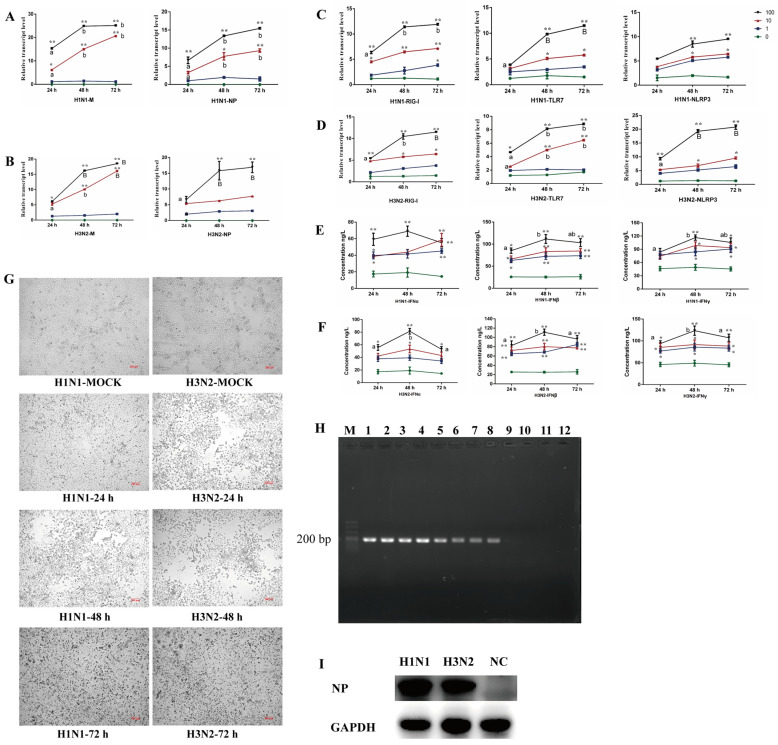
Establishment of alveolar macrophage cell line (3D4/21) models infected by H1N1 and H3N2. (**A**,**B**) represent the qPCR results of virus genome genes (*M* and *NP*) in 3D4/21 cells infected by H1N1 and H3N2. (**C**,**D**) represent the qPCR results of host genes (*RIG-I*, *TLR7* and *NLRP3*) in 3D4/21 cells infected by H1N1 and H3N2. (**E**,**F**) represent the ELISA results of virus-related cytokines in 3D4/21 cells infected by H1N1 and H3N2. (**G**) represents the microscope observation result of 3D4/21 cells infected by H1N1 and H3N2 (100-fold TCID_50_). (**H**) represents expression of *NP* gene in 3D4/21 cells infected by H1N1 and H3N2. (**I**) represents expression of NP protein in 3D4/21 cells infected by H1N1 and H3N2. For (**A**–**F**), * Indicates that there is a significant difference between different virus doses at the same time point (*p <* 0.05), ** indicates *p <* 0.01. Different lowercase letters indicate that the dose of the same virus is significantly different at different time points (*p <* 0.05), and different capital letters indicate *p <* 0.01. 0, 1, 10, and 100 represent 0-fold TCID_50_ (no virus infection), 1-fold TCID_50_, 10-fold TCID_50_ and 100-fold TCID_50_, respectively (For H1N1, 1-fold TCID_50_, 10-fold TCID_50_ and 100-fold TCID_50_ mean that 100 µL H1N1 virus diluted 10^4.7^, 10^3.7^, and 10^2.7^ times was used to infect 3D4/21 cells, respectively. For H3N2, 1-fold TCID_50_, 10-fold TCID_50_ and 100-fold TCID_50_ mean that 100 µL H3N2 virus diluted 10^2.75^, 10^1.75^, and 10^0.75^ times was used to infect 3D4/21 cells, respectively). For (**H**), M represents DL500 DNA marker. 1–4 stands for H1N1 group, 5–8 stands for H3N2 group, and 9–12 stands for non-infected (NC) group.

**Figure 2 ijms-23-01875-f002:**
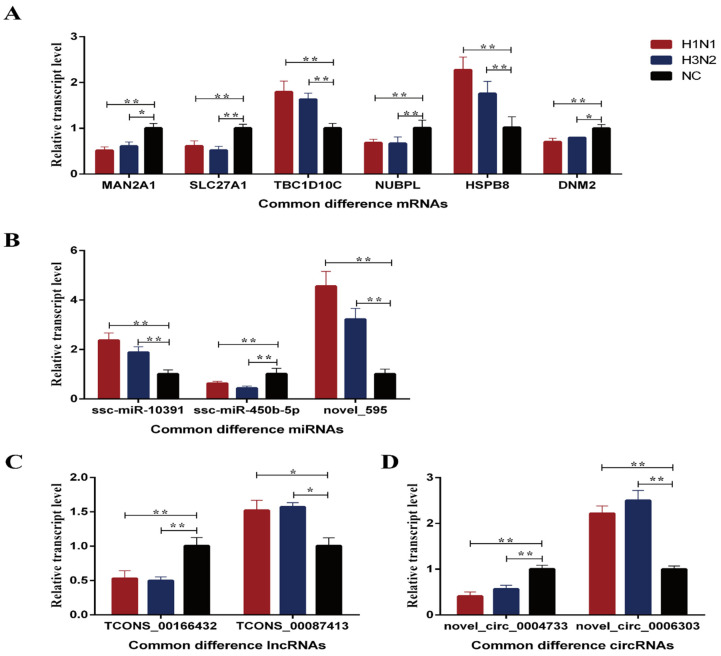
Fluorescence quantitative PCR results of partially differential RNAs. (**A**–**D**) represents differentially expressed mRNAs, miRNAs, lncRNAs and circRNAs, respectively. * indicates significant difference (*p <* 0.05), ** indicates significant difference (*p <* 0.01).

**Figure 3 ijms-23-01875-f003:**
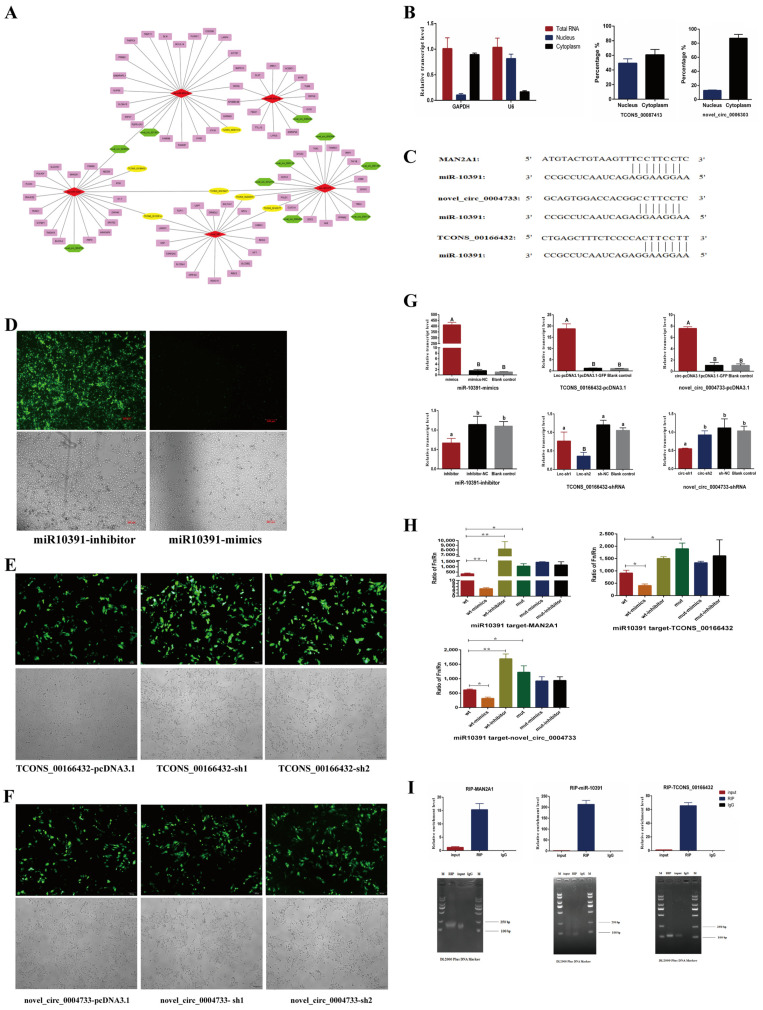
Verification of ceRNA networks in 3D4/21 cells infected by H1N1 and H3N2. (**A**) represents the predicted ceRNA networks diagram. The red diamond represents miRNA, the yellow ellipse represents lncRNA, the green hexagon represents circRNA, and the pink rectangle represents mRNA. (**B**) represents the expression of TCONS_00166432 and novel_circ_0004733 in nuclear and cytoplasmic of 3D4/21 cells. (**C**) represents the sequence alignment of targeted binding sites. (**D**–**F**) represent fluorescence expression of 3D4/21 cells transfected by recombinant vectors. (**G**) represents efficiency detection results of overexpression and interference. Different lowercase letters indicate significant differences (*p <* 0.05), and different capital letters indicate significant differences (*p <* 0.01). (**H**) represents the test result of dual luciferase activity. * indicates significant difference (*p <* 0.05), ** indicates significant difference (*p <* 0.01). The same below. (**I**) represents the results of RNA immunoprecipitation (RIP).

**Figure 4 ijms-23-01875-f004:**
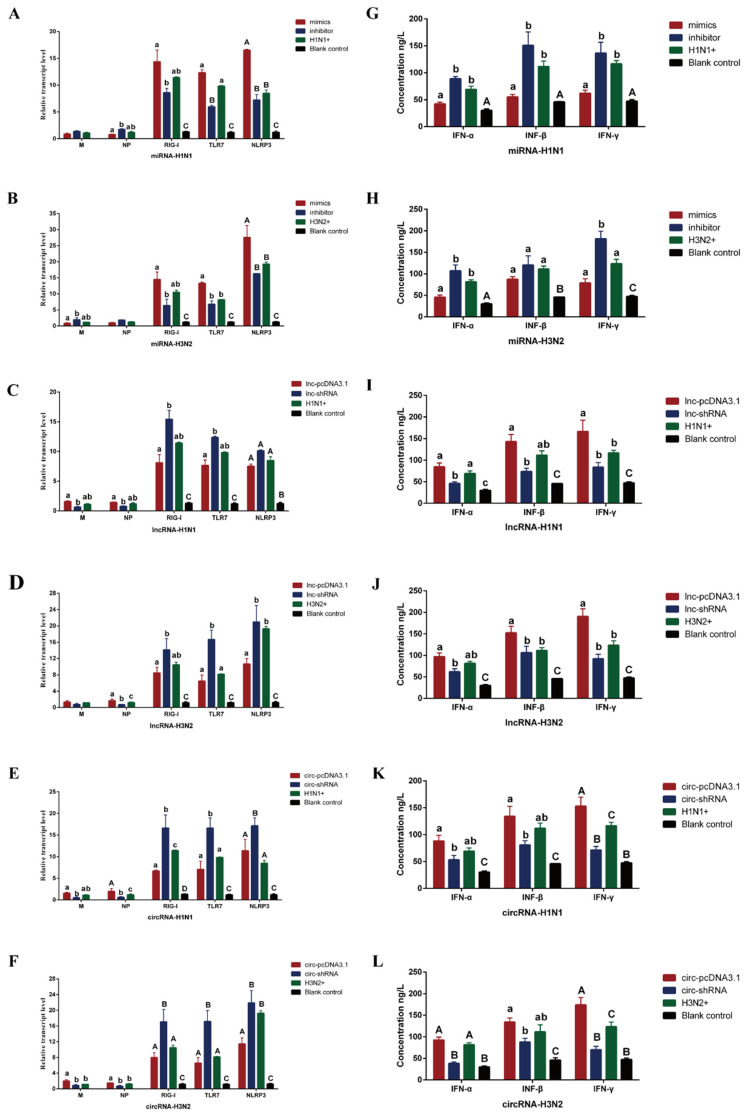
The regulatory role of miR-10391, TCONS_00166432 and novel_circ_0004733 in the process of H1N1 and H3N2 infecting 3D4/21 cells. (**A**–**F**) represents the effect of miR-10391, TCONS_00166432, and novel_circ_0004733 on the expression levels of viral genes and host genes in 3D4/21 cells infected by H1N1 and H3N2, respectively. (**G**–**L**) represents the effect of miR-10391, TCONS_00166432 and novel_circ_0004733 on the secretion of cytokines in the supernatant of 3D4/21 cells infected by H1N1 and H3N2, respectively. Different lowercase letters indicate significant differences (*p <* 0.05), and different capital letters indicate significant differences (*p <* 0.01).

**Figure 5 ijms-23-01875-f005:**
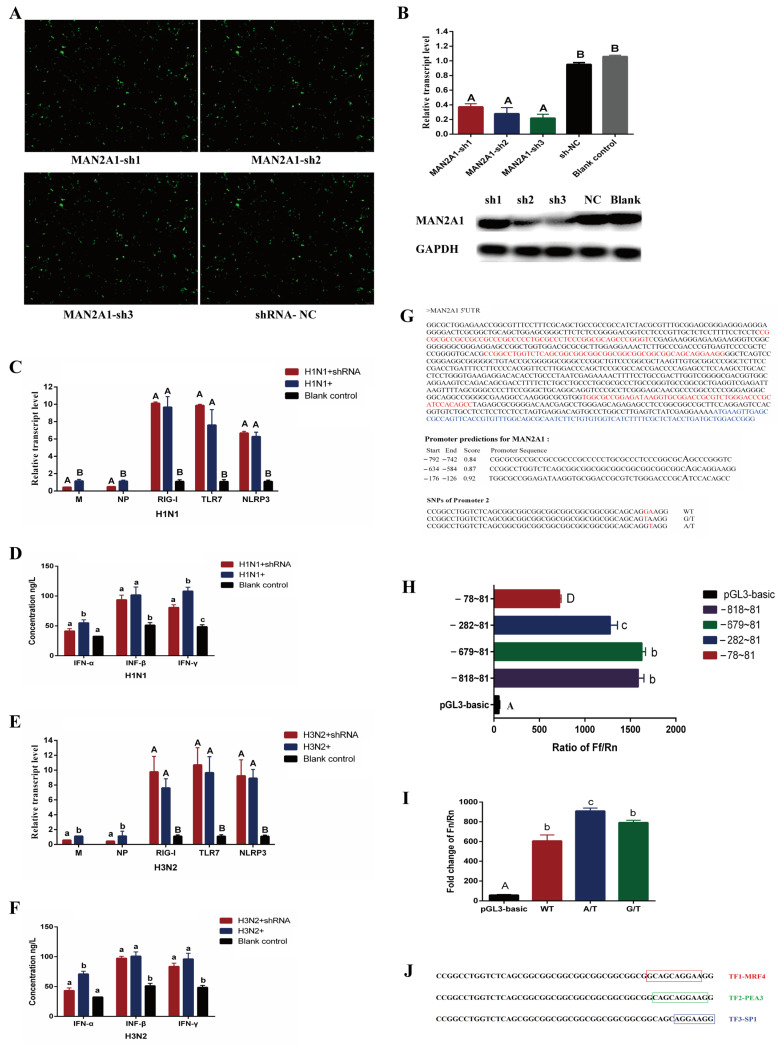
The regulatory role of *MAN2A1* gene in the process of H1N1 and H3N2 infecting 3D4/21 cells. (**A**) represents fluorescence expression of 3D4/21 cells transfected by *MAN2A1* interference vectors. (**B**) represents the qPCR and Western blot results of the *MAN2A*1 gene in 3D4/21 cells transfected by *MAN2A1* interference vectors. Different lowercase letters indicate significant differences (*p <* 0.05), and different capital letters indicate significant differences (*p <* 0.01). The same as below. (**C**,**E**) represent the effects of *MAN2A*1 gene on the expression levels of viral genes and host genes in 3D4/21 cells infected by H1N1 and H3N2. (**D**,**F**) represent the effects of *MAN2A*1 gene on the secretion of cytokines in the supernatant of 3D4/21 cells infected by H1N1 and H3N2. (**G**) represents prediction of *MAN2A1* gene promoter. The red letters represent the predicted core promoter sequence, and the blue letters represent the transcript sequence of the *MAN2A1* gene. (**H**) represents the result of dual luciferase detection of *MAN2A1* gene promoter. (**I**) represents the effect of *MAN2A1* gene promoter SNP on transcriptional activity. (**J**) represents the transcription factor bound by the SNP sequence in the core promoter region of *MAN2A1* gene. TF represents the abbreviation of transcription factor.

**Figure 6 ijms-23-01875-f006:**
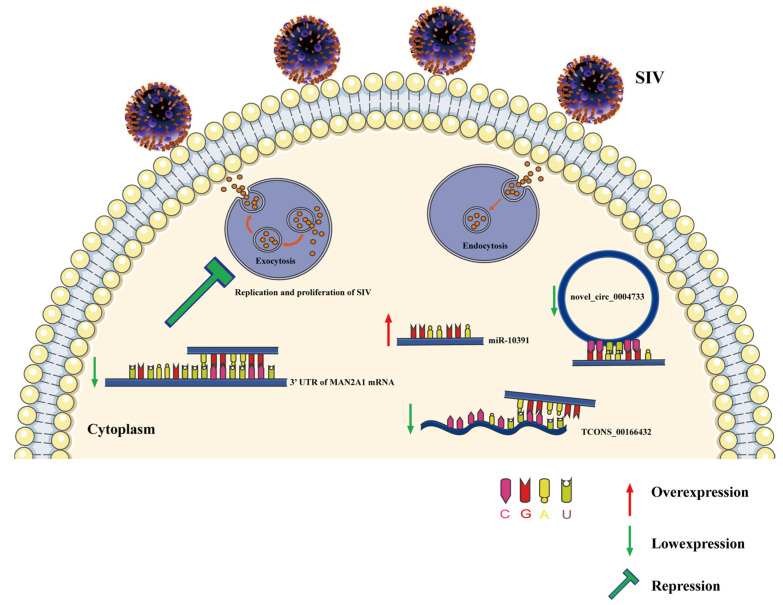
A schematic model displaying the role of TCONS_00166432/novel_circ_0004733-miR-10391-MAN2A1 in 3D4/21 cells infected by SIV.

**Table 1 ijms-23-01875-t001:** Partially common differential mRNAs, lncRNAs, circRNAs and miRNAs.

Name	log_2_ (Fold Change) (H1N1 VS NC)	q-Value (H1N1 VS NC)	log_2_ (Fold Change) (H3N2 VS NC)	q-Value (H3N2 VS NC)
MAN2A1	−8.2936	0.0420	−8.2641	0.0351
SLC27A1	−11.5320	0.0264	−11.7066	0.0181
TBC1D10C	11.0438	0.0391	11.5744	0.0131
NUBPL	−13.8219	0.0232	−14.0008	0.0169
HSPB8	11.3101	0.0412	12.3397	0.0454
DMN2	−16.8831	0.0334	−17.0719	0.0289
TRMT11	−13.0863	0.0378	−13.2738	0.0295
SH3GLB1	−12.6420	0.0076	−12.8302	0.0044
RNF4	−9.3709	0.0200	−10.7517	0.0066
SKAP2	−15.6092	0.0135	−15.7825	0.0102
TCONS_00166432	−14.0270	0.0212	−14.2238	0.0146
TCONS_00087413	9.0444	0.0195	9.0147	0.0148
TCONS_00034803	12.7180	0.0001	13.4636	0.0000
TCONS_00049633	13.3814	0.0255	13.1156	0.0032
TCONS_00049634	10.8614	0.0288	11.1368	0.0015
TCONS_00161069	−11.0298	0.0309	−11.2239	0.0219
TCONS_00073487	−12.1590	0.0144	−12.3543	0.0086
TCONS_00081028	14.2291	0.0020	14.5317	0.0109
TCONS_00318986	−14.5898	0.0144	−14.7840	0.0103
TCONS_00263471	−12.5471	0.0110	−12.7271	0.0064
novel_circ_0004733	−0.5734	0.0009	−0.4438	0.0036
novel_circ_0005622	−3.116	0.0159	−3.7254	0.0044
novel_circ_0006303	2.6657	0.0442	3.4964	0.0085
novel_circ_0006487	−1.2362	0.0029	−0.8881	0.0113
novel_circ_0006913	3.0359	0.0204	2.7151	0.0488
novel_circ_0008498	−1.2005	0.0095	−0.9471	0.0198
novel_circ_0008518	2.7173	0.0401	3.0313	0.0260
novel_circ_0012535	−2.7579	0.0361	−3.4083	0.0109
novel_circ_0012621	3.0912	0.0180	4.0189	0.0016
novel_circ_0012685	2.3074	0.0423	2.5896	0.0178
miR-375	0.7687	0.0071	0.7202	0.0006
miR-10391	2.6367	0.0071	1.9245	0.0016
novel_595	1.9539	0.0071	2.0738	0.0001
miR-450b-5p	−0.6574	0.0133	−0.8330	0.0021
miR-450c-5p	−0.9235	0.0004	−0.8091	0.0029

**Table 2 ijms-23-01875-t002:** Target gene prediction of partial differential lncRNAs (co-location).

Diff_lncRNA ID	Target_mRNA_Gene ID	Target_mRNA_Gene Name
TCONS_00166432	ENSSSCG00000014195	MAN2A1
TCONS_00087413	ENSSSCG00000026898	FAM149B1
TCONS_00261070	ENSSSCG00000001968	NUBPL
TCONS_00151394	ENSSSCG00000013630	DNM2
TCONS_00186867	ENSSSCG00000008498	HEATR5B

**Table 3 ijms-23-01875-t003:** Prediction of miRNA binding sites of partial differential circRNAs.

Diff_circRNA ID	miRNA ID	Energy
novel_circ_0004733	ssc-miR-10391	−18.66
novel_circ_0004733	ssc-miR-26b-5p	−16.10
novel_circ_0004733	ssc-miR-9847-3p	−17.74
novel_circ_0004733	ssc-miR-1296-5p	−16.46
novel_circ_0004733	ssc-miR-9798-3p	−10.71
novel_circ_0006303	ssc-miR-106a	−20.11
novel_circ_0006303	ssc-miR-122-5p	−18.42
novel_circ_0006303	ssc-miR-15b	−14.98
novel_circ_0006303	ssc-miR-20a-5p	−17.04
novel_circ_0006303	ssc-miR-195	−13.56

**Table 4 ijms-23-01875-t004:** Target genes prediction of 3 differential miRNAs.

Diff_miRNA ID	Target_mRNA_Gene ID	Target_Gene Name
miR-10391	ENSSSCT00000056127	MAN2A1
miR-10391	ENSSSCT00000010790	GTPBP1
miR-10391	ENSSSCT00000012079	POLR2F
miR-10391	ENSSSCT00000014885	PRPH
miR-10391	ENSSSCT00000015165	DNAJC22
miR-450b-5p	ENSSSCT00000000547	NUP50
miR-450b-5p	ENSSSCT00000010790	FGFR1OP2
miR-450b-5p	ENSSSCT00000051120	PABPC4
miR-450b-5p	ENSSSCT00000060293	TRMT11
miR-450b-5p	ENSSSCT00000065972	CCDC68
novel_595	ENSSSCT00000000818	RIBC2
novel_595	ENSSSCT00000002713	SULT4A1
novel_595	ENSSSCT00000025424	ORMDL2
novel_595	ENSSSCT00000032528	GPR162
novel_595	ENSSSCT00000046639	DSP

## Data Availability

The data presented in this study are available on request from the corresponding author.

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
