# Peer review of "The Competitive Endogenous RNA (ceRNA) Regulation in Porcine Alveolar Macrophages (3D4/21) Infected by Swine Influenza Virus (H1N1 and H3N2)"

_ijms, 2022, doi:10.3390/ijms23031875_

Round 1
Reviewer 1 Report
The authors argued that the competitive endogenous RNA regulates 3D4/21 cells against swine influenza viruses. The authors predicted the ceRNA network and investigated the amounts of interferons secreted into the cell supernatant. However, since the main target cells of swine influenza viruses are epithelium cells in lungs, it is quite skeptical that this study provide “a basis for the study of transcriptional regulation of swine influenza viruses” (page 2, line 75). Furthermore, the main problem of this manuscript is lacking introductions of ceRNA and ncRNA-miRNA-mRNA network, so that the readers hardly understand the potential merits of the study.
Major comments
- Throughout the manuscript (including the title), there are several typographical and grammatical errors. The authors should consider English proofreading the manuscript.
- The authors should explain more in detail on Introduction about ceRNA; otherwise, the readers cannot understand “a new mechanism for RNA interaction (page 2, line 59)”. Since the RNA interaction of ceRNA is a main topic in this manuscript, the authors need to explain briefly. Just siting a reference is not enough.
- The authors need to reconsider the definition of TCID50. It is impossible with the virus titer 10-4.7 TCID50/100μl (page 2, line 84). Furthermore, what is 1-fold TCID50, 10-fold TCID50, and 100-fold TCID50 (page 2, line 85)? These expressions must be revised.
- The strain names of H1N1 and H3N2 are missing. Since swine influenza viruses have many lineages, strains critically affect to the results.
- The figures are too small to read throughout the manuscript (Figure 1 A-G). The authors should rearrange the figures.
Minor comments
- Page 1, lines 43-44: “mixed containers” should be rephrased with “mixing vessels”.
- Page 2, line 57: the authors should spell out “miRNA, lncRNA, and circRNA”.
- The authors should describe in correct abbreviations. For example, “cytopathic conditions (CPE) (Page 2, line 82)” should be rephrased to “cytopathic effect (CPE)”.
- Page 2, line 84: Spell out “TCID50”
- Materials and methods: Cell culture condition for D4/21 cells should be stated.
Author Response
Comments and Suggestions for Authors:
The authors argued that the competitive endogenous RNA regulates 3D4/21 cells against swine influenza viruses. The authors predicted the ceRNA network and investigated the amounts of interferons secreted into the cell supernatant. However, since the main target cells of swine influenza viruses are epithelium cells in lungs, it is quite skeptical that this study provide “a basis for the study of transcriptional regulation of swine influenza viruses” (page 2, line 75). Furthermore, the main problem of this manuscript is lacking introductions of ceRNA and ncRNA-miRNA-mRNA network, so that the readers hardly understand the potential merits of the study.
Thank you for your comments. As we know, swine influenza viruses can multiply in the nasal mucosa, tonsils, trachea, bronchial lymph nodes and lungs. And the lung is the main target organ. The 3D4 /21 cell line is an in vitro passage cell line of porcine alveolar macrophages (PAM), which has lost its phagocytic function in vitro, but still retains the function of cytokine secretion. It has been reported as a cell model for studying the inflammatory response caused by porcine reproductive and respiratory syndrome virus (PRRSV), porcine circovirus type 2 (PCV2), mycoplasma pneumonia (M.hyo), swine influenza virus (SIV), streptococcus suis type 2 (SS2) and Haemophilus parasuis (H. parasuis). In this study, we found that Swine influenza viruses H1N1 and H3N2 could infect 3D4/21 cells, replicate and proliferate in host cells, and could cause changes in the expression of virus-related genes and changes in the secretion of cytokines. Based on the above research foundation, we chose 3D4/21 cells as the research model to explore the relevant regulatory mechanism of swine influenza virus infecting host cells. In addition, we added more details of ceRNA and ncRNA-miRNA-mRNA network in the revised manuscript (page 2, line 58-63). Finally, We have modified the introduction, methods and results according to your suggestions.
Su, Y.; Shi, P.; Zhang, L.; Lu, D.; Zhao, C.; Li, R.; Zhang, L.; Huang, J. The Superimposed Deubiquitination Effect of OTULIN and Porcine Reproductive and Respiratory Syndrome Virus (PRRSV) Nsp11 Promotes Multiplication of PRRSV. J. Viro. 2018, 92, e00175-18.
Wang, X.; Xu, X.; Wang, W.; Yu, Z.; Wen, L.; He, K.; Fan, H. MicroRNA-30a-5p promotes replication of porcine circovirus type 2 through enhancing autophagy by targeting 14-3-3. Arch. Virol. 2017,162, 2643-2654.
Zhang, Z.; Wei, Y.; Liu, B.; Wu, Y.; Wang, H., Xie, X.; Feng, Z.; Shao, G., Xiong, Q. Hsp90/Sec22b promotes unconventional secretion of mature-IL-1β through an autophagosomal carrier in porcine alveolar macrophages during Mycoplasma hyopneumoniae infection. Mol. Immunol. 2018, 101, 130-139.
Wang, Z.; Chai, W.; Burwinkel, M.; Twardziok, S.; Wrede, P.; Palissa, C.; Esch, B.; Schmidt, M.F. Inhibitory influence of Enterococcus faecium on the propagation of swine influenza A virus in vitro. PloS One 2013, 8, e53043.
Wang, W.; He, K.; Ni, Y.; Lv, L.; Zhou, J.; Zhang, X.; Yu, Z.; Mao, A.; Wen, L.; Wang, X.; Li, B.; Guo, R. mRNA level of inflammation-associated cytokines of porcine alveolar macrophages cell lines 3D4 /21 stimulated by Streptococcus suis serotype 2. Jiangsu J. of Agr. Sci. 2013, 29, 559-564. (in Chinese)
Shen, Y.; Zhou, N.; Zhang, J.; Wang, M.; Li, Y.; Jiang, P. TGF-β1 Signaling Stimulates Invasion of Haemophilus parasuis into 3D4/21 Cells. Acta Veterinaria et Zootechnica Sinica 2018, 49, 1720-1726. (in Chinese)
Major comments:
- Throughout the manuscript (including the title), there are several typographical and grammatical errors. The authors should consider English proofreading the manuscript.
Thanks for your comment. We modified the whole manuscript (including the title) to make it more reasonable. For specific changes, please check the revised manuscript.
- The authors should explain more in detail on Introduction about ceRNA; otherwise, the readers cannot understand “a new mechanism for RNA interaction (page 2, line 59)”. Since the RNA interaction of ceRNA is a main topic in this manuscript, the authors need to explain briefly. Just siting a reference is not enough.
Thank you for your suggestion. We have added more information of ceRNA in our revised manuscript (page 2, line 58-63): “It is known that microRNA can cause gene silencing by binding to mRNA, and ceRNA can regulate gene expression by competitively binding to microRNA. It was reported that lncRNAs and circRNAs regulated the expression of mRNA (with the same miRNA binding sites) by functioning as competing endogenous RNAs (miRNA sponge), namely lncRNA-miRNA-mRNA and circRNA-miRNA-mRNA regulation network [6-8]”.
Tay, Y.; Rinn, J.; Pandolfi, P.P. The multilayered complexity of ceRNA crosstalk and competition. Nature 2014, 505, 344–352.
Fan, C.N.; Ma, L.; Liu, N. Systematic analysis of lncRNA-miRNA-mRNA competing endogenous RNA network identifies four-lncRNA signature as a prognostic biomarker for breast cancer. J. Transl. Med. 2018, 16, 264.
Rengganaten. V.; Huang, C.J.; Tsai, P.H.; Wang, M.L.; Yang, Y.P.; Lan, Y.T.; Fang, W.L.; Soo, S.; Ong, H.T.; Cheong, S.K.; et al. Mapping a Circular RNA-microRNA-mRNA-Signaling Regulatory Axis That Modulates Stemness Properties of Cancer Stem Cell Populations in Colorectal Cancer Spheroid Cells. Int. J. Mol. Sci. 2020, 21, 7864.
- The authors need to reconsider the definition of TCID50. It is impossible with the virus titer 10-4.7TCID50/100μL(page 2, line 84). Furthermore, what is 1-fold TCID50, 10-fold TCID50, and 100-fold TCID50 (page 2, line 85)? These expressions must be revised.
Thank you for your comments. In this study, we detected the TCID50 of H1N1 and H3N2 use the MDCK cells. The TCID50 of H1N1 was 10-4.7/100 μL, which mean inoculation of 100 µL of H1N1 virus diluted 104.7 times can cause CPE in 50% of MDCK cells. The TCID50 of H3N2 was 10-2.75/100 μL, which mean that inoculation of 100 µL of H3N2 virus diluted 102.75 times can cause CPE in 50% of MDCK cells. For H1N1, 1-fold TCID50, 10-fold TCID50 and 100-fold TCID50 mean that 100 µL H1N1 virus diluted 104.7, 103.7, and 102.7 times was used to infect 3D4/21 cells respectively. For H3N2, 1-fold TCID50, 10-fold TCID50 and 100-fold TCID50 mean that 100 µL H3N2 virus diluted 102.75, 101.75, and 100.75 times was used to infect 3D4/21 cells respectively. To simplify the description, we use 1-fold TCID50, 10-fold TCID50 and 100-fold TCID50 to represent different virus doses. We have added corresponding instructions in the revised manuscript (legend of Figure 1).
- The strain names of H1N1 and H3N2 are missing. Since swine influenza viruses have many lineages, strains critically affect to the results.
Thank you for your comment. We have added the strains information in our revised manuscript (page 16, line 461-463): Virus strains “A/swine/Liaoning/32/2006 (H1N1) and A/swine/Heilongjiang/10/2007 (H3N2) were gifts from Professor Guoqiang Zhu, College of Veterinary Medicine, Yangzhou University”.
- 5. The figures are too small to read throughout the manuscript (Figure 1 A-G). The authors should rearrange the figures.
Thanks for your suggestion. We have modified the Figure 1 A-G.
Minor comments
- Page 1, lines 43-44: “mixed containers” should be rephrased with “mixing vessels”.
Thanks for your comment. We have modified the description of “mixed containers” with “mixing vessels” in our revised manuscript (Page 1 line 41).
- Page 2, line 57: the authors should spell out “miRNA, lncRNA, and circRNA”.
Thank you for your suggestion. We have modified the description with “four types of RNA-microRNA (miRNA), long non-coding RNA (lncRNA), messenger RNA (mRNA) and circular RNA (circRNA)” in the revised manuscript (Page 2, lines 55-57).
- The authors should describe in correct abbreviations. For example, “cytopathic conditions (CPE) (Page 2, line 82)” should be rephrased to “cytopathic effect (CPE)”.
Thank you for your advice. We have modified the description of “cytopathic conditions” with “cytopathic effect” in our revised manuscript (Page 2, line 83).
- Page 2, line 84: Spell out “TCID50”.
Thank you for your suggestion. We have added the full names of “TCID50” with “50% tissue culture infective dose” in the revised manuscript (Page 2, line 84).
- 5. Materials and methods: Cell culture condition for D4/21 cells should be stated.
Thanks for your comment. We have added the culture condition for 3D4/21 cells in Materials and methods part of our revised manuscript (Page 17, line 461-463): “3D4/21 cells were cultured with 1640 complete medium containing 10% fetal bovine serum in incubator under the condition of 37 ℃ and 5% CO2”.

Reviewer 2 Report
Swine influenza virus infection has been studied.Porcine alveolar macrophage cell (3D4/21) has been infected by swine influenza virus types of H1N1 and H3N2. Then, miRNAs, mRNAs, lncRNAs and circRNAs were checked. 2 ceRNAs (TCONS_00166432-miR10391-MAN2A1 and circ_0004733-miR10391-MAN2A1) were found to regulate H1N1 and H3N2 infected 3D4/21 cells.The approaches were performed using bioinformatic, expression, interference, qPCR, promoter luciferase and RNA immunoprecipitation (RIP). The their target gene MAN2A1 has been discovered by the interference, qPCR and antibodies. TCONS_00166432 and circ_0004733 bind to miR-10391 to target the MAN2A1, affecting SIV-infected 3D4/21 cells. In conclusion, the authors conclude ceRNA regulates the swine influenza virus infected 3D4/21 cells.
The study is interesting in the SI infection however, the present study is too premature to justify their claims.
1.The real mechanism how the ceRNA regulates their tatgets during SIV infection to porcine Macriphages.
2. Instead of macrophage cells, airway epithelial cells should be examined if this is for infection-based mechanism.
3. The used SIV should be typed by its sialic acid binding capacity in porcine cells. Is it infectious even for avians?
4. The miRNAs are diverse if the infection is progressed. More specific interaction between the RNAs and targets, and also multiple structure genes should be examined at the protein level. This will make clear how they are regulated.
Author Response
Comments and Suggestions for Authors:
Swine influenza virus infection has been studied. Porcine alveolar macrophage cell (3D4/21) has been infected by swine influenza virus types of H1N1 and H3N2. Then, miRNAs, mRNAs, lncRNAs and circRNAs were checked. 2 ceRNAs (TCONS_00166432-miR10391-MAN2A1 and circ_0004733-miR10391-MAN2A1) were found to regulate H1N1 and H3N2 infected 3D4/21 cells. The approaches were performed using bioinformatic, expression, interference, qPCR, promoter luciferase and RNA immunoprecipitation (RIP). The their target gene MAN2A1 has been discovered by the interference, qPCR and antibodies. TCONS_00166432 and circ_0004733 bind to miR-10391 to target the MAN2A1, affecting SIV-infected 3D4/21 cells. In conclusion, the authors conclude ceRNA regulates the swine influenza virus infected 3D4/21 cells.
The study is interesting in the SI infection. However, the present study is too premature to justify their claims.
Thank you for your comments. We have modified the introduction, methods and results according to your suggestions. For specific changes, please check the revised manuscript.
- The real mechanism how the ceRNA regulates their targets during SIV infection to porcine Macrophages.
Thanks for your comment. In this study, we illustrated that TCONS_00166432 and novel_circ_0004733 acted as miRNA sponges to regulate the expression of miR-10391, which targeted the MAN2A1 gene and regulated the replication and proliferation of SIV or immune responses in 3D4/21 cells. We have added a mechanism diagram in the manuscript to further illustrate the regulatory relationship (Figure 6. A schematic model displaying the role of TCONS_00166432/novel_circ_0004733-miR-10391-MAN2A1 in 3D4/21 cells infected by SIV).
- Instead of macrophage cells, airway epithelial cells should be examined if this is for infection-based mechanism.
Thank you for your comment. As we know, swine influenza viruses can multiply in the nasal mucosa, tonsils, trachea, bronchial lymph nodes and lungs. And the lung is the main target organ. The 3D4 /21 cell line is an in vitro passage cell line of porcine alveolar macrophages (PAM), which has lost its phagocytic function in vitro, but still retains the function of cytokine secretion. It has been reported as a model cell for studying the inflammatory response caused by porcine reproductive and respiratory syndrome virus (PRRSV), porcine circovirus type 2 (PCV2), mycoplasma pneumonia (M.hyo), swine influenza virus (SIV), streptococcus suis type 2 (SS2) and Haemophilus parasuis (H. parasuis). In this study, we found that Swine influenza viruses H1N1 and H3N2 could infect 3D4/21 cells, replicate and proliferate in host cells, and could cause changes in the expression of virus-related genes and changes in the secretion of cytokines. Based on the above research foundation, we chose 3D4/21 cells as the research model to explore the relevant regulatory mechanism of swine influenza virus infecting host cells. And our results supported our conclusions.
Su, Y.; Shi, P.; Zhang, L.; Lu, D.; Zhao, C.; Li, R.; Zhang, L.; Huang, J. The Superimposed Deubiquitination Effect of OTULIN and Porcine Reproductive and Respiratory Syndrome Virus (PRRSV) Nsp11 Promotes Multiplication of PRRSV. J. Viro. 2018, 92, e00175-18.
Wang, X.; Xu, X.; Wang, W.; Yu, Z.; Wen, L.; He, K.; Fan, H. MicroRNA-30a-5p promotes replication of porcine circovirus type 2 through enhancing autophagy by targeting 14-3-3. Arch. Virol. 2017,162, 2643-2654.
Zhang, Z.; Wei, Y.; Liu, B.; Wu, Y.; Wang, H., Xie, X.; Feng, Z.; Shao, G., Xiong, Q. Hsp90/Sec22b promotes unconventional secretion of mature-IL-1β through an autophagosomal carrier in porcine alveolar macrophages during Mycoplasma hyopneumoniae infection. Mol. Immunol. 2018, 101, 130-139.
Wang, Z.; Chai, W.; Burwinkel, M.; Twardziok, S.; Wrede, P.; Palissa, C.; Esch, B.; Schmidt, M.F. Inhibitory influence of Enterococcus faecium on the propagation of swine influenza A virus in vitro. PloS One 2013, 8, e53043.
Wang, W.; He, K.; Ni, Y.; Lv, L.; Zhou, J.; Zhang, X.; Yu, Z.; Mao, A.; Wen, L.; Wang, X.; Li, B.; Guo, R. mRNA level of inflammation-associated cytokines of porcine alveolar macrophages cell lines 3D4 /21 stimulated by Streptococcus suis serotype 2. Jiangsu J. of Agr. Sci. 2013, 29, 559-564. (in Chinese)
Shen, Y.; Zhou, N.; Zhang, J.; Wang, M.; Li, Y.; Jiang, P. TGF-β1 Signaling Stimulates Invasion of Haemophilus parasuis into 3D4/21 Cells. Acta Veterinaria et Zootechnica Sinica 2018, 49, 1720-1726. (in Chinese)
- The used SIV should be typed by its sialic acid binding capacity in porcine cells. Is it infectious even for avians?
Thank you for your comment. The virus strains used in this study were A/swine/Liaoning/32/2006(H1N1) and A/swine/Heilongjiang/10/2007(H3N2). We have added the strains information in our revised manuscript (page 16, line 453-455). In Lu’s study, 4 of the H1N1 virus genes are homologous to the swine influenza virus genes, and 6 of the H3N2 virus genes are homologous to the swine influenza virus genes. There is currently no evidence weather the SIV in this study is infectious for avians.
Lu W. Isolation and identification of swine influenza virus H1N1 and H3N2 subtypes and development of swine influenza bivalent vaccine, inactivated (H1N1+H3N2). [D]. Yangzhou University, 2019. DOI:10.27441/d.cnki.gyzdu.2019.001826. (in Chinese)
- The miRNAs are diverse if the infection is progressed. More specific interaction between the RNAs and targets, and also multiple structure genes should be examined at the protein level. This will make clear how they are regulated.
Thank you for your comments. Indeed, the expression changes of a variety of miRNAs are involved in the course of infection. In this study, 5 miRNAs that were co-differentially expressed in 3D4/21 cells infected by H1N1 and H3N2 were screened through sequencing analysis (Figure S1), and the total number of target genes was predicted to more than 200 (data not shown in the manuscript). However, we mainly focus on the ceRNA regulatory relationship during the infection process in this study. Therefore, we only focused on the targeting relationship between this miRNA (miR-10391) and the MAN2A1 gene (which is also an mRNA that was differentially expressed after H1N1 and H3N2 infected 3D4/21 cells). The target and regulatory function analysis of other miRNAs remain to be revealed in our future research.

Round 2
Reviewer 2 Report
The authors have revised and responded to the previous criticisms.
However, the major and critical questions are not substantially revised with additional data.
For general science, the competitive endogenous RNA (ceRNA)-interacting/binding sequences should be illustrated. Just RNAs Just regulation are not accepted for the precise mechanism to explain.
Raw cell issue is also questionable but agreeable, although airway epithelial cells are desired.
Author Response
Response to Reviewer 2 Comments
Point 1: The authors have revised and responded to the previous criticisms. However, the major and critical questions are not substantially revised with additional data. For general science, the competitive endogenous RNA (ceRNA)-interacting/binding sequences should be illustrated. Just RNAs Just regulation are not accepted for the precise mechanism to explain.
Response 2: Thanks for your suggestion. We illustrated the binding sequences of ceRNAs in Figure3C. The seed sequences of miR-10391 completely matched the 3’ UTR sequences of target gene MAN2A1. In addition, miR-10391 had 7 base matched with novel_circ_0004733 and TCONS_00166432, respectively. What’s more, we verified their targeting relationships through dual luciferase activity assay and RIP test. Therefore, we illustrated that miR-10391 could inhibit the expression of MAN2A1 gene by binding the 3’UTR of MAN2A1 gene, while novel_circ_0004733 and TCONS_00166432 could act as the sponge of miR-10391 to reduce the effect of miR-10391 on MAN2A1 gene by binding the sequences of miR-10391. We have modified the description in our revised manuscript (page 7-8, line 180-183-315; page 14, line 315-318; Figure 6).
Figure 3. Verification of ceRNA networks in 3D4/21 cells infected by H1N1 and H3N2. A represents the predicted ceRNA networks diagram. The red diamond represents miRNA, the yellow ellipse represents lncRNA, the green hexagon represents circRNA, and the pink rectangle represents mRNA. B represents the expression of TCONS_00166432 and novel_circ_0004733 in nuclear and cytoplasmic of 3D4/21 cells. C represents the sequence alignment of targeted binding sites. D-F represent fluorescence expression of 3D4/21 cells transfected by recombinant vectors. G represents efficiency detection results of overexpression and interference. Different lowercase letters indicate significant differences (P <0.05), and different capital letters indicate significant differences (P <0.01). H represents the test result of dual luciferase activity. * indicates significant difference (P <0.05), ** indicates significant difference (P <0.01). The same below. I represents the results of RNA immunoprecipitation (RIP).
Figure 6. A schematic model displaying the role of TCONS_00166432/novel_circ_0004733-miR-10391-MAN2A1 in 3D4/21 cells infected by SIV
Point 2: Raw cell issue is also questionable but agreeable, although airway epithelial cells are desired.
Response 2: Thanks for your comment and agreement. Please contact us if there is anything we can do to improve our manuscript.

Round 3
Reviewer 2 Report
The authors have revised but the cells-related issues are not completely solved in the revision, only answering to the question.
Author Response
Point 1: The authors have revised but the cells-related issues are not completely solved in the revision, only answering to the question.
Response 1: Thanks for your comments. We explained why we used 3D4/21 cells in our study, and you agreed with our answer in the last revision. Of course, We agree with you very much that airway epithelial cells are an ideal model to study the mechanism of swine influenza virus infection. But it is unrealistic to replace 3D4/21 cells with it for all research experiments in the current manuscript. We will follow your advice and conduct experiments with airway epithelial cells in future swine influenza virus infection studies. Thanks again for your guidance.
